# Proton Beam Therapy in the Oligometastatic/Oligorecurrent Setting: Is There a Role? A Literature Review

**DOI:** 10.3390/cancers15092489

**Published:** 2023-04-26

**Authors:** Simona Gaito, Giulia Marvaso, Ramon Ortiz, Adrian Crellin, Marianne C. Aznar, Daniel J. Indelicato, Shermaine Pan, Gillian Whitfield, Filippo Alongi, Barbara Alicja Jereczek-Fossa, Neil Burnet, Michelle P. Li, Bethany Rothwell, Ed Smith, Rovel J. Colaco

**Affiliations:** 1Proton Clinical Outcomes Unit, The Christie NHS Proton Beam Therapy Centre, Manchester M20 4BX, UK; 2Division of Clinical Cancer Science, School of Medical Sciences, The University of Manchester, Manchester M13 9PL, UK; 3Department of Oncology and Hemato-Oncology, University of Milan, 20122 Milan, Italy; 4Department of Radiation Oncology, IEO European Institute of Oncology IRCCS, 20126 Milan, Italy; 5Department of Radiation Oncology, University of California, San Francisco, CA 94720, USA; 6National Lead Proton Beam Therapy NHSe, Manchester M20 4BX, UK; 7Department of Radiation Oncology, University of Florida, Jacksonville, FL 32206, USA; 8Department of Proton Beam Therapy, The Christie Proton Beam Therapy Centre, Manchester M20 3DA, UK; 9Advanced Radiation Oncology Department, IRCCS Ospedale Sacro Cuore don Calabria, 37024 Verona, Italy; 10Division of Radiology and Radiotherapy, University of Brescia, 25121 Brescia, Italy; 11Department of Radiation Oncology, Peter MacCallum Cancer Centre, Melbourne, VIC 3000, Australia; 12Sir Peter MacCallum Department of Oncology, The University of Melbourne, Melbourne, VIC 3010, Australia; 13Division of Physics, Department of Radiation Oncology, Massachusetts General Hospital and Harvard Medical School, Boston, MA 02114, USA

**Keywords:** oligometastases, oligorecurrences, proton beam therapy

## Abstract

**Simple Summary:**

Stereotactic ablative radiotherapy (SABR) and stereotactic radiosurgery (SRS) with conventional photon radiotherapy (XRT) are well-established treatment options for oligorecurrent/oligometastatic disease. Here, we review the available evidence surrounding the current use of proton beam therapy (PBT) in this setting. We describe how the particular physical properties of PBT could be used for the treatment of oligometastases/oligorecurences. Moreover, we further outline how current research has the potential to expand the therapeutic window of PBT whilst minimising the intrinsic uncertainties of this technique. This would potentially lead to an expansion of the commissioning of PBT to include these indications.

**Abstract:**

Background: Stereotactic ablative radiotherapy (SABR) and stereotactic radiosurgery (SRS) with conventional photon radiotherapy (XRT) are well-established treatment options for selected patients with oligometastatic/oligorecurrent disease. The use of PBT for SABR-SRS is attractive given the property of a lack of exit dose. The aim of this review is to evaluate the role and current utilisation of PBT in the oligometastatic/oligorecurrent setting. Methods: Using Medline and Embase, a comprehensive literature review was conducted following the PICO (Patients, Intervention, Comparison, and Outcomes) criteria, which returned 83 records. After screening, 16 records were deemed to be relevant and included in the review. Results: Six of the sixteen records analysed originated in Japan, six in the USA, and four in Europe. The focus was oligometastatic disease in 12, oligorecurrence in 3, and both in 1. Most of the studies analysed (12/16) were retrospective cohorts or case reports, two were phase II clinical trials, one was a literature review, and one study discussed the pros and cons of PBT in these settings. The studies presented in this review included a total of 925 patients. The metastatic sites analysed in these articles were the liver (4/16), lungs (3/16), thoracic lymph nodes (2/16), bone (2/16), brain (1/16), pelvis (1/16), and various sites in 2/16. Conclusions: PBT could represent an option for the treatment of oligometastatic/oligorecurrent disease in patients with a low metastatic burden. Nevertheless, due to its limited availability, PBT has traditionally been funded for selected tumour indications that are defined as curable. The availability of new systemic therapies has widened this definition. This, together with the exponential growth of PBT capacity worldwide, will potentially redefine its commissioning to include selected patients with oligometastatic/oligorecurrent disease. To date, PBT has been used with encouraging results for the treatment of liver metastases. However, PBT could be an option in those cases in which the reduced radiation exposure to normal tissues leads to a clinically significant reduction in treatment-related toxicities.

## 1. Introduction

The concept of “oligometastatic disease” was introduced in 1995 by Hellman and Weichselbaum [1] to describe those clinical situations in which the site and number of metastases are limited to five or fewer [2].

The definition of “oligorecurrent disease” was clarified by Niibe et al. [3] to indicate those clinical scenarios in which one or few recurrences (usually one) are present in one or few organs. The disease recurrence can occur in the same or in a different organ to the primary. Theoretically, the main difference between oligometastatic and oligorecurrent disease lies in the uncontrolled or controlled nature of the primary lesion. Oligorecurrence requires a controlled primary lesion, whereas in oligometastatic disease the primary lesion can be either uncontrolled or controlled. These definitions have evolved and been clarified over the years, as follows [4,5]:-Synchronous oligometastatic disease: includes patients found to have metastatic disease at the time of initial diagnosis.-Metachronous oligometastatic disease (often used interchangeably with oligorecurrence): refers to patients initially treated with definitive therapy to cure their malignancy who subsequently (>3 months later) develop limited disease recurrence.-Oligoprogressive disease: represents patients with known metastatic disease who exhibit few isolated areas of progression in a background of otherwise stable disease.-Oligopersistent disease: persistent disease after systemic therapy.

Traditionally, the treatment intent in both metastatic and recurrent settings was considered “palliative” rather than curative—aimed at symptom relief rather than disease eradication. However, the improvement of systemic cytotoxic and molecularly targeted therapies has allowed for the sterilisation of micrometastases and a paradigm shift in cancer care. Theoretically, if one or a few gross metastatic/recurrent sites could be eradicated by local therapy, these patients could be cured with concomitant systemic treatments. Therefore, even though substantial differences exist in prognosis that are mainly dependent on the nature of the primary lesion, oligometastatic/oligorecurrent diseases are generally considered to be amenable to curative local treatments, such as surgery or radiation therapy. Previous results suggest that radiation therapy could induce a more robust immune response and better harness the synergy of radiotherapy and antitumor immunity compared to surgery [6]. In effect, especially in combination with systemic therapy, the treatment of oligometastatic/oligorecurrent tumours can achieve “disease modification”, which is of value to the patient.

In patients with good performance status and no contraindications from other comorbidities, metastasectomy is usually considered to be the gold standard [7,8]. SRS and SABR have emerged relatively recently as treatment options for patients with cranial and extracranial oligometastatic disease, respectively [9,10]. Local control (LC) of the metastatic sites is proven to slow down further disease progression and, in selected patient groups, potentially improve the overall survival (OS) [11,12]. Moreover, SRS and SABR are non-invasive outpatient procedures with very low morbidity, are highly tolerable in frail patients, and avoid the removal of functional tissue. SRS/SABR can also allow for the treatment of multiple lesions on the same day, minimising treatment interruptions for patients receiving systemic therapy [13].

Proton beam therapy (PBT), with its inherent physical properties of the Bragg peak and the lack of an exit dose, is apt for a number of clinical indications [14]. Nevertheless, due to its high cost and limited availability, national health policies in the UK and EU have supported the routine commissioning of PBT for selected tumour indications that are considered to be curable [15,16,17]. However, due to dramatic reductions in the cost of PBT equipment, PBT has undergone exponential growth in demand and capacity worldwide [18]. The increase in the number of proton treatment rooms available worldwide will potentially lead to an expansion of its commissioning to include indications that are currently not routinely funded. The aim of this review is to evaluate the role and current utilisation of PBT in the oligometastatic/oligorecurrent settings.

## 2. Materials and Methods

To identify relevant information, the research question was framed according to the PICO criteria (Population, Intervention, Comparison, and Outcomes) [19]. The population of the study was cancer patients with (oligo)recurrence OR (oligo)metastases OR solitary metastasis. The intervention was the radiation treatment with proton therapy OR stereotactic body proton therapy (SBPT) OR radiosurgery with proton therapy (SRS with proton therapy). There was no specific comparison for this study, as the comparison with conventional photon radiotherapy was not within the scope of this study. The outcomes of this study focused on the clinical outcomes of PBT in the oligorecurrent/oligometastatic settings.

Using Medline and Embase, a comprehensive literature review was conducted in December 2022, using the following terms:

(((“proton therap*” OR “proton beam therap*” OR protontherap* OR “proton beam radiation therap*” OR “proton radiotherap*”).mp OR Proton Therapy/) OR (“Stereotactic Body Proton Therapy “ OR sbpt OR (srs and proton*) OR (stereotactic and proton*)).mp.) AND (oligometast* OR “solitary recurrence” OR “oligo recurrence”).mp.

The process of obtaining the relevant papers for this study is shown in the PRISMA diagram (Figure 1).

## 3. Results

The study characteristics are listed in Table 1. Among the selected articles, 6/16 records originated in Japan, 6 in the USA, and 4 in Europe. The focus was oligometastatic disease in 12, oligorecurrence in 3, and both oligorecurrent and oligometastatic disease in 1. Most of the studies analysed (9/16) were retrospective cohorts, one was a case series, two were case reports, two were phase II clinical trials, one was a literature review, and one study discussed the pros and cons of PBT in the oligometastatic/oligorecurrent settings. From the studies presented, a total of 925 patients are included in this review.

The following paragraphs focus on organ-specific oligometastases and oligorecurrence, as these are discussed in the included studies:

### 3.1. Various Anatomical Sites

Rans et al. [22] used stereotactic body proton therapy (SBPT) to retrospectively replan 25 lesions in various body sites from 12 castration-resistant prostate cancer patients previously treated with photon stereotactic ablative radiotherapy (XRT SABR). For planning comparison purposes, the same treatment planning system and plan objectives (i.e., 99% of the prescribed dose to 99% of the volume) were applied. There were no significant differences in the plan quality metrics (i.e., target dose coverage, homogeneity index, conformity index, gradient index) between the SBRT and SBPT plans. However, improved sparing of the organs at risk (OARs) was made possible with the SBPT plans, with the only exception being the dose to the skin, which was significantly higher with SBPT and is a known problem with PBT [23]—albeit one that can usually be abrogated by careful planning. The extent to which this dosimetric advantage translates into a clinical benefit is unclear and is currently being investigated in many settings through normal tissue complication probability modelling [24,25,26].

Bakhtiar et al. [27] analysed the outcomes of 299 patients in the USA who received PBT within the final 12 months of their life. Nearly half of them (45%) were treated with PBT for recurrent disease in the primary site, while 16% were treated for isolated recurrences elsewhere or oligometastatic disease. Even though many of these patients received treatment with definitive PBT doses and concurrent systemic therapy, the incidence of grade 3 acute toxicity was acceptable (15%), without G > 3 acute toxicities. However, on average, the patients spent 24% of their remaining days on treatment (the median time from the final fraction to death was 139 days (1–363 days)). Thus, the authors suggest the incorporation of a prognostic indicator to further optimise the use of PBT in the end-of-life setting.

### 3.2. Lungs and Thoracic Lymph Nodes

Contreras et al. [28] retrospectively evaluated the outcomes of 25 patients re-irradiated with radical doses of PBT (median 60 Gy, range 40–62.5) for thoracic recurrences or metachronous malignancies (mainly from non-small-cell lung cancer (NSCLC)). Their reported one-year local control (LC) was 74.8%, and they concluded that PBT re-irradiation is a viable treatment option for patients with thoracic recurrences or metachronous malignancies, offering excellent outcomes in patients with limited therapeutic options. However, given the potential risks of acute and late toxicity, patient selection is crucial to maximise the benefits of therapy.

The role of particle therapy in the treatment of oligometastatic lung disease was reported by Sulaiman et al. [29] on a cohort of 47 patients treated for 59 lung metastases from various primary sites (26 lesions treated with PBT and 33 with carbon ions). The 2-year LC rate was 79%, without significant differences in LC rates between carbon ion therapy and PBT according to univariate analysis. The cumulative rate of grade 3 acute/late toxicity was 13%. Specifically, the local control was lower in lung metastases from colorectal cancers but improved significantly when the prescription dose was escalated to a BED_10_ of ≥110 GyE_10_, irrespective of the primary tumour site. These findings were later confirmed by Aibe et al. [30] on a retrospective series of 118 patients with lung oligometastases. Therefore, both groups concluded that particle therapy is effective and well tolerated in the context of oligometastatic lung tumours.

Irradiation of thoracic lymph node oligometastases with PBT has been reported in a case report describing the irradiation of a solitary metachronous pre-tracheal lymph-node in a breast cancer patient [31], and in a retrospective cohort of 33 patients treated for lymph node oligorecurrences from NSCLC (17 of whom were treated with PBT and 16 with XRT) [32]. In the latter series, the PBT subgroup received a median dose of 70 GyRBE (60–76) in conventional fractionation, with local progression-free rates approaching 80% at 3 years, but with no statistically significant (*p* = 0.084) improvement compared to XRT. Although not strictly SABR, this provides further data to support the use of radiotherapy in the treatment of oligorecurrences.

### 3.3. Liver

Nakajima et al. [33] reported the use of image-guided (IG)-PBT in 43 patients with 53 lesions treated for oligometastatic liver disease from gastric/colorectal cancer (GCRC). IG-PBT with hypofractionated regimens (66 GyRBE in 10 fractions to a peripherally located tumour, and 72.6 GyRBE in 22 fractions to a centrally located tumour) appeared to be effective for the local control of liver oligometastases from GCRC, particularly in the subgroup of patients with a shorter history of systemic therapy. 

The interim results of a phase I/II trial of 60 GyRBE in three fractions with SBPT suggest excellent local control outcomes not only with minimal toxicity, but also affording the option of subsequent courses for out-of-field liver recurrences [34]. In their literature review, Gill et al. [35] referenced a phase II clinical trial of 89 patients with liver metastases of varying histological types (mostly colorectal carcinomas (CRCs)) [36] as a proof-of-concept to show that PBT is remarkably well tolerated (i.e., no grade ≥ 3 toxicities reported) and effective even for oligometastases that are 6 cm or larger (the 1- and 3-year local control rates were 71.9% and 61.2%, respectively). In line with previous photon series, the strongest predictor of poor LC was mutation in KRAS, particularly when also associated with TP53 mutation [37].

### 3.4. Bone

Ishikawa et al. [38] described the successful treatment of a solitary metastasis of the sternum—a site with limited tolerance due to previous photon irradiation of the left chest for primary treatment of the left breast. The patient was prescribed 70 GyRBE in 2.5 GyRBE per fraction and was reported to be in complete remission at 3 years, without receiving any systemic treatment. Similar conclusions were drawn by Johnson et al. [39] in a case series of four patients treated for solitary metastases of the sternum in women previously treated with XRT for cancer of the left breast. In the clinical scenarios presented in this article, dosimetric comparisons showed comprehensive coverage of the metastatic site and optimal sparing of the heart with pencil-beam scanning PBT as compared to XRT techniques (3D, three dimensional; VMAT, volumetric modulated arc therapy), with decreased Dmean to the heart, to the left anterior descending artery, and to both lungs.

### 3.5. Pelvis

Chuter et al. [20] retrospectively planned 10 pelvic cancer recurrences previously treated with XRT SABR, to test whether SBPT could be an option in those cases where meeting OAR constraints can be challenging. Dosimetric comparison of the relevant OAR statistics showed a decrease in OAR dose using SBPT over SABR in all patients, with equivalent target coverage. The largest statistically significant reduction was seen for the colon, followed by the small bowel, sacral plexus, and cauda equina. However, the magnitude of the clinical benefit of these OAR reductions remains unclear.

### 3.6. Brain

In a single-institution retrospective analysis of 815 metastases from 370 patients treated with PBT SRS between 1991 and 2016, Atkins et al. [21] reported local control and toxicity outcomes comparable to those obtained with conventional photon SRS strategies. They concluded that, although PBT SRS remains resource-intensive, future strategies should focus on selecting those patients who would benefit most from integral dose reduction.

**Table 1 cancers-15-02489-t001:** List of the 16 studies presented in the review. Abbreviations: Mets. = metastases. Oligomets:Oligometastases. N/A Not applicable; Nil: nothing; OS: overall survival; LC: local control; AT: acute toxicities; LT: late toxicities; * these studies report numbers of toxicity events rather than toxicity incidence rates; ◊ these are retrospective comparative planning studies.

N.	Author	Year	Country	Primary Tumour	Site of Mets	No. of	Setting	Article Type	Type of Study	Total Dose (GyRBE)	No. of Fractions	Dose per Fraction (GyRBE)	Local Control/Overall Survival Rates	Acute and Late Toxicities
						Patients/Lesions		1. Abstract						
								2. Article						
1	Rans et al. [22]	2022	Belgium	Prostate	Various	12/25	Oligomets	1	Retrospective cohort ◊	36		12	N/A	N/A
2	Nakajima et al. [33]	2019	Japan	Gastric/colorectal	Liver	43/53	Oligomets	1	Retrospective cohort	66 or 72.6		6.6	1- and 2- year OS: 87% and 63%, respectively; 1- and 2-year LC: 73% and 70%, respectively	N/A
3	Hoyer et al. [40]	2018	Denmark	N/A	N/A		Oligomets	1	Descriptive abstract *	N/A		N/A	N/A	N/A
4	Contreras et al. [28]	2017	USA	Various	Lung	25 patients	Recurrence	1	Retrospective cohort	Median 60 (40–62.5)		Median 2 (2–10)	1-year OS: 82%; 1-year LC: 74.8%,	AT: 12% (G > 3); LT: 20% (G > 3)
5	Sufficool et al. [34]	2018	USA	Various	Liver	06/09	Oligomets	1	Phase II trial	60		20	Median follow-up 9.8 months (1–33): LC 100%	AT: 1 patient G1 fatigue
6	Bakhtiar et al. [27]	2021	USA	Various	Various	182 patients	Oligomets/recurrence	1	Retrospective cohort	Median 50 (15–80)		2	Median OS: 139 days (1–363)	AT: 85% (any grade); LT: 17% (any grade)
7	Sulaiman et al. [29]	2014	Japan	Various	Lung	47/59	Oligomets	2	Retrospective cohort	Median 60 (52.8–70.2)	Median 8 (4–26)		1- and 2-year OS: 72.7% and 54%, respectively; 1- and 2-year LC: 88.4% and 79%, respectively	AT *: G1 23, G2 1, G3 2; LT *: G1 19, G2 6, G3 4
8	Johnson et al. [38]	2022	USA	Breast	Sternum	4 patients	Oligomets	2	Case series ◊	60 (sternum); 45–50.4 (other targets)		2–2.4 (sternum); 1.8–2 (other targets)	Median follow-up 28 months: LC 100%	AT *:5 G2
9	Aibe et al. [30]	2021	Japan	Various	Lung	118/141	Oligomets	2	Retrospective cohort	Median 64 (52.8–89.6)	Median 10 (4–40)	6.6 (2–13.2)	1- and 2-year OS: 79% and 67.8%, respectively; 1- and 2-year LC: 92.2% and 86.3%, respectively	AT: 7% (G ≥ 2); LT: 8% (G2)
10	Ishikawa et al. [38]	2022	Japan	Breast	Sternum	01/01	Oligomets	2	Case report	70		2.5	NED at 3-years follow-up	AT: 1 G2
11	Kawamata et al. [31]	2020	Japan	Breast	Lymph nodes	01/01	Oligomets	2	Case report	60		2	NED at last follow-up	Nil
12	Gill et al. [35]	2018	UK	Colorectal	Liver		Oligomets	2	Review	N/A		N/A	N/A	N/A
13	Nakamura et al. [32]	2020	Japan	Lung	Lymph nodes	33 patients	Recurrence	2	Retrospective cohort	Median 70 (66–76)		2	3-year OS: 63.8%; 3-year LC: 79.7%,	AT: 11 patients G2 (33%), 1 G3 (3%); LT: 1 G3 (3%)
14	Hong T et al. [36]	2017	USA	Various	Liver	89 patients	Oligomets	2	Phase II trial	Median 40 (30–50)		Median 8 (6–10)	Median OS: 18.1 months; 1- and 3-year LC: 71.9% and 61.2%, respectively	AT: 87.6% G2 ≤ 2
15	Atkins et al. [21]	2018	USA	Various	Brain	370/815	Oligomets	2	Retrospective cohort	Median 18 (8–28)	1		Median follow-up 9.2 months; 6- and 12-month local failure: 4.3% and 8.5%, respectively; 6- and 12-month OS: 76.0% and 51.5%, respectively	LT: 3.6% radionecrosis at 12 months
16	Chuter et al. [20]	2022	UK	Various	Pelvis	10/10	Recurrence	2	Retrospective cohort ◊	30	5	6	N/A	N/A

## 4. Discussion

Stereotactic radiotherapy with XRT offers an excellent, minimally invasive ablative treatment option for the treatment of metastases of the brain (SRS) and body (SABR). SRS and SABR are also proven to increase the LC in radioresistant tumours [40,41] and can provide an overall survival advantage in patients with good prognostic factors (e.g., young age, good performance status, primary tumour control) [12,42,43,44]. These techniques can avoid the surgical morbidity related to metastasectomy and are usually well tolerated [45,46]. International guidelines have been established that elucidate the best treatment modality with regards to total dose and dose per fraction, in consideration of patient and disease characteristics (e.g., primary disease, close proximity to organs at risk with serial architecture, volume and site of the lesion, treatment aim) [47,48,49].

The physical and biological characteristics of PBT allow significant reductions in the Dmean doses to normal tissues surrounding the target volume, without compromising its coverage, although in many situations the clinical advantage (to the patient) in terms of reduced side effects may be only modest. This is commonly illustrated with the “Bragg peak” and the depth–dose curve for PBT, which most clearly represents its properties. PBT presents a slightly stronger biological effect than XRT, as implied by a generic relative biological effectiveness (RBE) of 1.1. However, this “conversion factor” varies throughout the beam path, increasing at the distal end and fall-off regions [14]. Similar to XRT, different modes of PBT delivery exist. Passive scattering (PS-PBT) techniques use mechanical devices (e.g., collimators and compensators) in the particle trajectory to shape the beam to the tumour volume, including the depth, analogous to XRT’s step-and-shoot. However, when compared to advanced IMRT technology, the dose-conforming potential of PS-PBT is limited, mainly due to technical reasons [50]. In the newer pencil-beam scanning (PBS-PBT) technique, the beam is scanned across the target volume, and the energy is changed to allow conformation at depth and modulated in intensity at the source, which is similar to intensity-modulated photon techniques. Whilst PBS-PBT allows better conformity to the target, concerns have been raised that RBE uncertainties could be higher with PBS-PBT as compared to PS-PBT, in relation to different linear energy transfer (LET) profiles [51]. Moreover, PBS-PBT can also be particularly sensitive to organ motion [52]. 

Among the studies presented in this review, four are comparative studies [20,22,32,39]. Of these, three are retrospective comparative planning studies [20,22,39] in which the SBPT plans show a similar target dose coverage, with better sparing of the OARs (with the exception of the skin). The fourth is the only “real-life” comparative study of patients treated with either XRT or PBT for lung oligorecurrences [32]. This shows—with the limitation of the small patient cohort—no significant differences in acute and late toxicities but improved 3-year LC rates (68.8% vs. 90.9%, *p* = 0.054). One possible reason for this, according to the authors, is the better target coverage allowed by PBT as compared to XRT, albeit differences in background patient characteristics could not be excluded, and larger cohorts are needed to validate these results.

A major limitation of most comparative planning studies of PBT vs. XRT is their retrospective nature, with patients having been treated with one modality (most often XRT) and retrospectively planned for comparison. Reported practice in the comparison of XRT and PBT plans has been disparate, and the difference in uncertainties between the two modalities has frequently been neglected, making the comparison potentially biased. A literature review conducted by Lowe et al. highlighted that very few studies report how the uncertainty scenarios impact on CTV and PTV coverage, whilst no study has detailed how the uncertainty scenarios were applied to the assessment of OARs’ volumes [53].

The discussed technical features of PBT, together with its higher cost and limited availability, have made PBT less appealing for the treatment of metastases and prevented its wide adoption in this setting. However, major advances have been made on both levels, and future research in this field is promising. Treatment uncertainties are currently incorporated into the treatment planning margins around the target volume, and the use of proton mini-beams is a future prospect [54]. Moreover, due to dramatic reductions in the cost of PBT equipment, together with growth in demand, PBT has undergone exponential growth in capacity worldwide, with a possible expansion of its indications [11].

One unquestionable advantage of PBT is the reduction in the integral dose in all clinical scenarios [52]. For this reason, with current technologies and capacity, the more resource-demanding SBPT can be considered an option when it can offer a reduced risk of morbidity and late effects. Examples include specific re-irradiation settings in extracranial sites with parallel architecture, e.g., the liver, lungs, and some oligometastatic sites in paediatric patients and teenagers/young adults [55], where there is a rationale to prioritise the reduction in the integral dose. For organs with serial architecture, such as the brain, the concern over an increased risk of toxicity due to RBE uncertainty hinders the implementation of brain SRS with PBT at present [40]. Similar observations can be drawn for spinal metastases, where the key limiting factor in PBT planning is the effect of range uncertainty, which proved significant in a comparative XRT-PBT study of seven patients treated with XRT SABR and replanned with PBS-PBT. Even though PBT plans feature superior sparing of the OARs situated anteriorly, along with a lower maximum dose and higher conformity of the prescription dose to the target, the effect of range uncertainty is nevertheless significant, and this could result in poorer target coverage and higher doses to the spinal cord and cauda equina [56] (Figure 2).

With this in mind, it becomes clearer to understand why most of the studies presented in this review include patients treated for “extra-axial” oligometastases/oligorecurrences, where the priority is either given to the preservation of the remaining organ (with parallel architecture, i.e., liver, lung) or to reducing late complications for in-field recurrences of previously irradiated tissues. The median dose per fraction in the studies presented was 4.15 Gy RBE (2–20), with standard or moderately hypofractionated regimens being the most common. 

The reluctance to use extreme hypofractionated regimens in the oligometastatic/oligorecurrent setting is therefore partially due to RBE and range uncertainties, as well as the high sensitivity of PBS-PBT to organ motion, which might be “less forgiving” with high doses per fraction [57,58,59]. RBE varies with linear energy transfer (LET). In general, the more densely ionising the radiation, the more biologically effective it is. The biological effectiveness of a certain type of radiation depends on several factors, including LET, dose per fraction, dose rate, the presence or absence of oxygen, and the biological system or cell type involved. All of these factors can alter the dose–response relationship. As a consequence, for moderately hypofractionated regimens, the RBE is expected to drop below 1.1 [59], potentially leading to the danger of an underdosage in the tumour region, but not threatening normal tissues. Moreover, the end-of-beam fall-off (range uncertainty) depends, among other factors, on the dose per fraction; therefore, the magnitude of this shift could change. In the radical setting of early-stage NSCLC, evidence suggests that hypofractionated PBT is safe and effective when uncertainties are managed with advanced techniques (e.g., respiratory motion gating, robust optimisation) [60].

One more setback for the implementation of SBPT and PBT-SRS is the substantial difference in the use of image guidance between XRT and PBT. In the treatment room, cone-beam CT is currently used in many centres, but developments have been slow compared with other forms of radiotherapy. The dual-energy scanner is a particular method that is currently being investigated to determine stopping power more accurately and reduce range uncertainties [61].

Currently, a number of technical and biological advances open up new options for PBT, potentially expanding its use in this setting in the future. However, at present, in consideration of the unique challenges presented by PBT that are not seen in XRT, PBT has very limited indications in this setting—mainly limited to the treatment of large lesions in parallel organs, in which RBE and range uncertainties are less of an issue and the reduction in the integral dose is clinically meaningful. For instance, the need to decrease the integral dose to normal liver tissue is critical in patients with liver metastases in the context of underlying cirrhosis. There is evidence that normal liver function is significantly correlated with the percentage of normal liver tissue that is not irradiated. In these clinical scenarios, reducing the integral dose to the remaining liver allowed by PBT may help to preserve liver function, while also allowing for future retreatment of the liver [62]. In the brain SRS setting, the recent development of spot-scanning proton arc (SPArc) therapy may help mitigate RBE, as well as range and motion uncertainties [63]. Furthermore, recent evidence suggests a dosimetric advantage in single brain metastases utilising SPArc compared to VMAT [64]. Based on NTCP models, the patients with brain metastases who may benefit the most from SPArc could be those with large targets for which the reduction in the integral dose would lower the risk of radionecrosis.

An innovative form of PBT that is currently an area of active research in the field is pMBRT (Figure 3) [65]. pMBRT is a technique that combines spatial fractionation of the dose with submillimetric proton beams that are mainly created by mechanical collimators. That is, the dose distributions are characterised by alternating regions of high (peaks) and low (valleys) dosage. This dose pattern leads to a reduction in the toxicity to normal tissues [66], without compromising the rates of tumour control [67], due to the potential differential vascular, bystander-like, and immunomodulatory effects, as well as cell migration [65,66,67,68]. This increased normal tissue dose tolerance may contribute to diminishing the radiation-induced toxicity to tissues surrounding the target volume in the treatment of bulky metastases compared with stereotactic XRT. Indeed, the treatment planning study by Ortiz et al. [54] showed that pMBRT could lead to similar levels of target coverage and a reduction in the dose to OARs compared to stereotactic XRT in the treatment of various metastatic sites (i.e., brain, lung, and liver), while meeting recommended dose–volume constraints. pMBRT treatments were proposed to be delivered in one fraction and using 1–2 fields, which may lead to more favourable treatment regimes in terms of reduced costs and inter-fraction uncertainties. In fact, in preclinical studies where the whole brains of rats were irradiated, the motor (i.e., motor coordination, muscular tonus, and locomotor activity), emotional (i.e., anxiety, fear, motivation, and impulsivity), and cognitive (i.e., learning, memory, temporal processing, and decision-making) functions were preserved. These results suggest that pMBRT may enable the irradiation of large volumes of brain tissue in the treatment of patients with brain metastases, along with prophylaxis for microscopic disease, with minimal impact on the normal brain tissue at the functional level [69].

Furthermore, recent findings on FLASH irradiations with ultra-high dose rates could make hypofractionation a clinically advantageous strategy [70,71]. Preclinical studies have suggested that a normal-tissue-sparing effect could be achieved for irradiation at a sufficiently high dose rate (>40 Gy/s) [72], and with a dose per beam above about 5 Gy [49,73,74,75]. On the other hand, a so-called “FLASH effect” has in some cases been observed at doses as low as 4 Gy [76]. Although the requirements for a FLASH effect remain to be fully elucidated, SABR currently looks to be the most promising modality for FLASH delivery in a clinical setting, where PBT provides a viable route to its widespread clinical use [42]. However, any trade-off in fractionation scheduling to meet the requirements for FLASH should be carefully considered while many biological factors remain unclear [77].

## 5. Conclusions

In the setting of metastasis/recurrence-directed therapy, for patients with limited disease burden and a controlled primary lesion, PBT represents a feasible option. Patient selection remains crucial [78], with good performance status and the availability of systemic treatments representing the ideal clinical scenario. With current technologies, PBT can be considered in some specific settings—e.g., when reducing the whole-organ dose is a priority—to decrease morbidity and late effects. For the time being, small metastases in close relationship to serial organs might still benefit from photon IMRT techniques. However, comparative effectiveness studies between PBT and XRT in this setting are lacking. Current research and advances in technology will potentially expand the current paradigms.

## Figures and Tables

**Figure 1 cancers-15-02489-f001:**
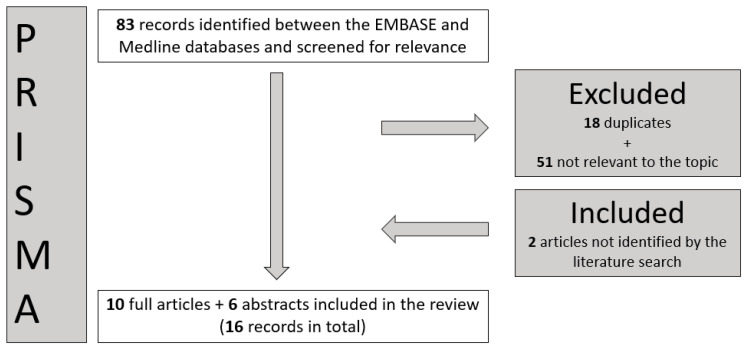
A Preferred Reporting Items for Systematic Reviews and Meta-Analysis (PRISMA) flowchart. The identification stage resulted in 83 records between Medline and Embase as of December 2022. Eighteen duplicates were excluded at this stage, and fifty-one were excluded after screening as irrelevant. Two further articles—not returned by the literature search—were identified by the authors and added [20,21]. These were not returned by the literature search as they did not contain the keywords used for the PICO selection.

**Figure 2 cancers-15-02489-f002:**
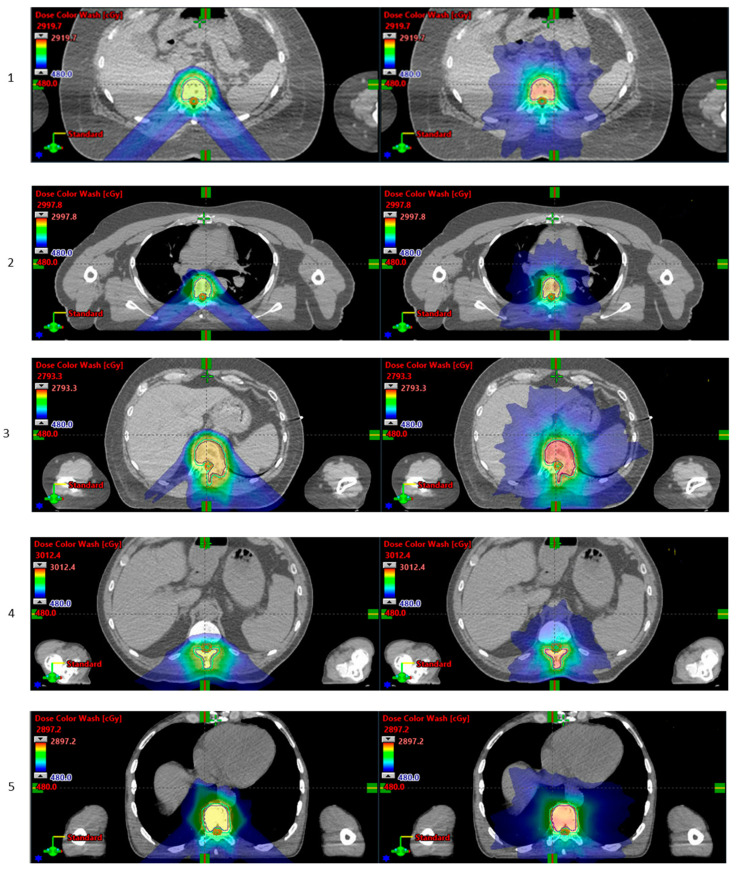
Plan comparison for 5 patients with spinal metastases: The contours of the clinical and planning target volumes (CTV and PTV) are shown, as well as the spinal cord/cauda equina and planning organs-at-risk volumes (PRVs). Proton plans are shown on the left column, and photon plans are shown on the right. The prescription doses are 24 Gy in 3 fractions. Doses between the 20% (4.8 Gy) and photon *D*_*max* are shown in colour wash (Courtesy of Matthew Lowe and Rovel Colaco, the Christie NHS PBT Centre).

**Figure 3 cancers-15-02489-f003:**
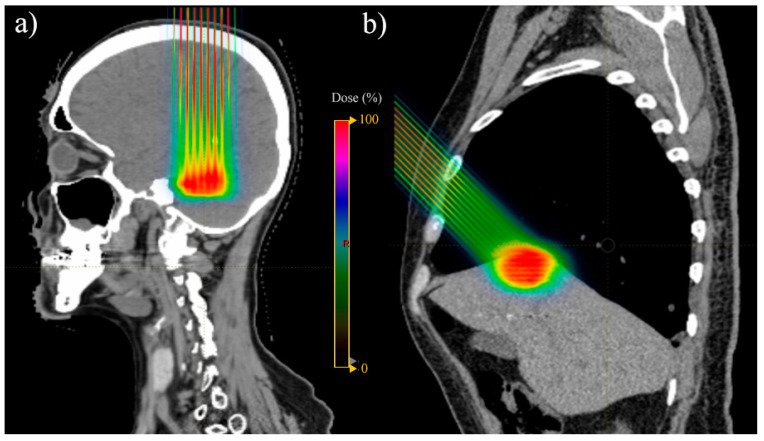
Examples of proton mini-beam radiotherapy (pMBRT) dose distributions in the treatment of a (**a**) brain and (**b**) liver metastases.

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
