# Peer review of "Proton Beam Therapy in the Oligometastatic/Oligorecurrent Setting: Is There a Role? A Literature Review"

_cancers, 2023, doi:10.3390/cancers15092489_

Round 1

Reviewer 1 Report

The cancers journal invited me to review the manuscript „Proton Beam Therapy in the oligometastatic/oligorecurrent setting: is there a role? A literature review“. A literature search was conducted with terms matching those of the special edition of cancers.

Generally the paper is well written and concise. However, I discerned the following major deficiencies:

·         1. In section 1 the clinical use is motivated with the favorable dose distribution of protons vs. photons. However, in section 4 the authors concede that the lateral dose fall-off of protons delivered with PBS deteriorates the dose distribution making it inferior to photon-based stereotactic machines for small lesions. As the expansion of proton therapy is strongly linked to PBS, the use of protons for stereotactic cases is highly questionable. This leaves protons for bigger metastases (l. 201, p 5) or for recurrent cases. Thus, from a technical point of view the stereotactic irradiations should be separated from the one of non-small recurrences and ones with conventional fractionation. The latter categories are, however, daily clinical practice in proton therapy centers and the authors should discuss the scientific aspects in the scope of the current study.

·         2. When investigating the use of protons for oligo***, the conclusions of the community about the use proton for stereotactical applications (in general) should be considered. This should be part of the discussion. Proton therapy is lagging the development of photon machines regarding image guidance and the corresponding integration with 3rd-party vendors. This is a concern especially for SABR and adds to the lateral penumbra and range uncertainty problems.

·         3. Regarding above item, see for example, J Appl Clin Med Phys. 2022;23:e13701, 10.1002/acm2.13701. The manuscript under review discusses FLASH and grid therapy regarding upcoming special techniques. That reference, however, discusses PBS with collimators and arc therapy as key enablers for stereotactical use. Although I personally doubt that arc therapy is a game changer for this use, the technical options discussed in the manuscript under review is clearly not complete. He-Ions are another “exotic” option.

·        4. The type of study should distinguish between planning studies and those studies in which patients were treated with protons (e.g. in Tab. 1).

Minor issues:

·         It seems that Ref. 50 is publically not available

Author Response

Thanks to the reviewer for his/her careful review of the manuscript and feedback. His/her comments have been taken onboard and clarified in the manuscript. I believe the suggested revision has improved the manuscript’s quality and has certainly been educational for the writer. I hereby provide a point-by-point response to the comments:

 (please see the attachment)

Reviewer 2 Report

This is a review article on proton beam therapy (PBT) for oligometastatic/oligorecurrent tumors. Since PBT has not been so often employed for these conditions, this article included only 16 reports dealing with various anatomical sites. So, the review is not an extensive one. However, due to the paucity of such articles, this paper may be of moderate value. There are many typos and the authors should be more careful in preparing their manuscript.  

Specific comments

L83: You mention surgery and radiation therapy as local therapy. Can you comment on the negative effect of surgery against host immunity, compared to PBT?

L131: Table 1 is missing.

L141 & 189: “K.” (before et al.) should be removed.

L156: Is “G>3” not “G≥ 3”?

L163: Remove )).

L184, 191, 195, and elsewhere: Gy and GyE should be GyRBE.

In Discussion, you discussed various aspects of PBT including the pencil beam scanning (PBS) technique but failed to discuss the increased RBE at the distal end of the beams (may be called distal end enhancement according to Nomura et al., doi.org/10.1080/09553002.2021.1889704). This effect seems to be more important in PBS PBT than in passive scattering PBT, because the biological dose distribution may change (become higher) in the target, as well as in the normal tissue just behind the target. This issue should be included in discussion.

L256: You mentioned the technical difficulty to generate narrow pencil beams. Currently, 4-6 mm beams are available, but do you mean still narrower beams are desirable?

L264-265: You should explain the “improved 3-year LC rates” in more detail. Why were the rates improved?

L304-306: The uncertainty in RBE especially at higher doses per fraction should be explained in more details. Why does RBE drop below 1.1 in moderately hypofractionated treatment?

L316~: Proton mini beam therapy may not be familiar to many readers, and should be explained in more details. What is the method/mechanism of producing mini beams and why is the toxicity in normal tissues reduced without compromising tumor control rates?

The usage of capital letters is questionable. Why do you capitalize many words like Proton Beam Therapy and others?

A space is missing after . (L89, L101 and elsewhere) and before [ ] (Line 89 and elsewhere). 

References should be provided following the journal style.

Author Response

Thanks to the reviewer for his/her careful review of the manuscript and feedback. His/her comments have been taken onboard and clarified in the manuscript. I believe the suggested revision has improved the manuscript’s quality and has certainly been educational for the writer. I hereby provide a point-by-point response to the comments:

 (please see the attachment, note that references are reported in the main manuscript)

Reviewer 3 Report

The purpose of this paper is to provide a literature review of proton beam therapy for oligometastatic/oligorecurrent disease.  There also seems to be an agenda to justify the use of protons in such a setting, however the evidence is not conclusive.  The authors show that protons are a feasible modality for these cases, but the cited literature do not demonstrate a clear advantage especially in light of the limited availability and higher cost.  Their conclusion, that protons "could represent an option" is the strongest statement that can be made.  Overall the paper is well written and explores an interesting option for proton therapy.  Some specific comments follow, by line number.

25: Stereotactic not Stereotactive

27: particular might be better than peculiar.  Their physical properties are what they are and no more peculiar than those of photons, at least to a physicist.

35: property is better than "advantage".  The word advantage is often used by the proton marketing department, but unless you are showing how that property provides an advantage I would refrain from using it. It looks biased.

42: One of the studies included was a literature review.  Did that review include any of the other literature from your own review? I'm concerned about double-counting.

97: "more favourable" also seems to be more of a marketing term than a scientific one.

98: "better sparing" is mentioned throughout the paper.  In reality OAR DVHs for comparative plans will cross, showing a proton advantage at low doses but a disadvantage at high doses.  I would caution against making this claim without being more specific as to what "better sparing" means.

118: In your literature search did you consider the term "hadrontherapy" or "ion therapy"? These might have yielded additional results.

128: it might be good to say which two studies were identified by the authors and not by the search as well as why they were not identified by your search (in the interest of full disclosure to avoid the appearance of bias).

141: Planning studies such as those by Rans et al. can be very misleading.  The photon plan has limitations: time constraints in the clinic being one.  Additionally, planning systems stop optimizing once goals are met, which is not the same as reducing OAR doses to the absolute minimum.  Most photon plans that are used in the clinic can be improved if pushed harder.  Comparing these to proton plans that are done after the fact can be misleading.  Additionally, such proton plans are often not practical. Proton margins may need to be larger which reduces some of their advantage. Clinically planners may use more beams than practical or violate other real-world clinical constraints when creating a plan for publication.  These are limitations of that study, which then become limitations of your paper.  You should include some of these limitations in the discussion.

147: Another mention of "improved sparing". Ideally sparing means reduction of morbidity.  Since that is not what you mean, you should be more explicit (e.g. reduction of tissue irradiated, or lower mean dose, etc.)

221: "decrease in OAR dose". In my experience max OAR dose is not reduced by proton beams. So please specify what dose is decreased.

223: Indeed, the clinical benefit is unclear.

228: "comparable" is good in that we are not saying "advantage".

230: Reduced integral dose is the advantage protons have. You need to identify when that will be a benefit: large target volumes, multiple mets, sensitive structures (e.g. lung V5).

243: "reductions in doses". Perhaps if you define up front what doses protons reduce you can then use that as your definition througout.

246: "modest" if at all.

253: Delete "continuously". In fact, more PBS systems are actually spot scanning providing a multitude of discrete pencil beams, not a continuously scanned pencil beam.

304: While biological effect is an issue, it is probably more the effect of hypofractionation than proton RBE.  Photon SRS doses may only be 1/3 the dose used for conventional fractionation whereas proton RBE is only a 10-20% correction typically.  Additional issues may be technological. PBS is more sensitive to motion, but over 30 fractions we have more confidence that it will average out than over 3-5 fractions.  Additionally, many existing proton centers to not have volumetric imaging for patient treatments which is generally required for SABR.

Author Response

Thanks to the reviewer for his/her careful review of the manuscript and feedback. His/her comments have been taken onboard and clarified in the manuscript. I believe the suggested revision has improved the manuscript’s quality and has certainly been educational for the writer. I hereby provide a point-by-point response to the comments:

 (please see the attachment)

Please note that references are reported in the main manuscript

Round 2

Reviewer 1 Report

The cancers journal invited me to review the revised manuscript „Proton Beam Therapy in the oligometastatic/oligorecurrent setting: is there a role? A literature review“. A literature search was conducted with terms matching those of the special edition of cancers. The authors clearly improved the paper. I think there is still potential for some improvements.

Abstract (see item 1 in previous iteration): I agree that liver metastasis are probably one of the few clinical application cases of stereotactic proton plans. The the fact that the study identified only some niches for potential beneficial use of protons in stereotactic RT is not represented accordingly in the abstract.

Line 338 (item 2 in previous iteration): I don’t the the meaning of the sentence “In fact, in treatment planning there is additional difficulty in using computed tomography (CT) images in dose calculation algorithms for PBT. “ I’m only aware of the stopping-power/range-uncertainty problem. But that one is addressed further below in the same paragraph.

Line 360, (item 3 in previous iteration): I disagree with the selection of advanced proton techniques, which might boost stereotactical proton therapy in future. pMBRT still needs translation into clinics. I personally see some physical limits, which might hinder clinical use for many indications. As mentioned before, SPAarc does not solve the issue of the bad conformality at high dose levels. PBS with apertures (MLCs, static, dynamic) have been used clinically and a lot of research is going on in this field.

Author Response

The cancers journal invited me to review the revised manuscript „Proton Beam Therapy in the oligometastatic/oligorecurrent setting: is there a role? A literature review“. A literature search was conducted with terms matching those of the special edition of cancers. The authors clearly improved the paper. I think there is still potential for some improvements.

Abstract (see item 1 in previous iteration): I agree that liver metastasis are probably one of the few clinical application cases of stereotactic proton plans. The the fact that the study identified only some niches for potential beneficial use of protons in stereotactic RT is not represented accordingly in the abstract.

This has now been added in the abstract to reflect this finding from analysed studies

Line 338 (item 2 in previous iteration): I don’t the the meaning of the sentence “In fact, in treatment planning there is additional difficulty in using computed tomography (CT) images in dose calculation algorithms for PBT. “ I’m only aware of the stopping-power/range-uncertainty problem. But that one is addressed further below in the same paragraph.

This is what I meant (stopping power-range uncertainty) but I understand that that line comes across as redundant and therefore has been removed.

Line 360, (item 3 in previous iteration): I disagree with the selection of advanced proton techniques, which might boost stereotactical proton therapy in future. pMBRT still needs translation into clinics. I personally see some physical limits, which might hinder clinical use for many indications. As mentioned before, SPAarc does not solve the issue of the bad conformality at high dose levels. PBS with apertures (MLCs, static, dynamic) have been used clinically and a lot of research is going on in this field.

This has been nuanced in the text. These have been described as areas of active research in the field.
